# Water and Nitrogen Coupling Increased the Water-Nitrogen Use Efficiency of Oilseed Flax

**DOI:** 10.3390/plants12010051

**Published:** 2022-12-22

**Authors:** Zhengjun Cui, Zechariah Effah, Bin Yan, Yuhong Gao, Bing Wu, Yifan Wang, Peng Xu, Haidi Wang, Bangqing Zhao, Yingze Wang

**Affiliations:** 1State Key Laboratory of Arid Land Crop Science, Lanzhou 730070, China; 2College of Agronomy, Gansu Agricultural University, Lanzhou 730070, China; 3CSIR-Plant Genetic Resources Research Institute, Bunso 999064, Ghana; 4College of Life Science and Technology, Gansu Agricultural University, Lanzhou 730070, China

**Keywords:** coupling irrigation and nitrogen, oilseed flax, grain yield, water- and nitrogen- use efficiency

## Abstract

Increasing water shortages and environmental pollution from excess chemical nitrogen fertilizer use necessitate the development of irrigation-nitrogen conservation technology in oilseed flax production. Therefore, a two-year split-plot design experiment (2017–2018) was conducted with three types of irrigation (I) levels (no irrigation (I0), irrigation of 1200 m^3^ ha^−1^ (I1200), and 1800 m^3^ ha^−1^ (I1800)) as the main plot and three nitrogen (N) application rates (0 (N0), 60 (N60) and 120 (N120) kg N ha^−1^) as the subplot in Northwest China to determine the effects of irrigation and N rates on oilseed flax grain yield, yield components, water-use efficiency (WUE), and N partial factor productivity (NPFP). The results show that I1800 optimized the farmland water storage and water storage efficiency (WSE), which gave rise to greater above-ground biomass. Under I1800, the effective capsule (EC) number increased significantly with increasing irrigation amounts, which increased significantly with increasing nitrogen application rate (0–120 kg ha^−1^). Both irrigation and nitrogen indirectly affect GY by affecting EC; the highest grain yield was observed at the I1800N60 treatment, which increased by 69.04% and 22.80% in 2017 and 2018 compared with the I0N0 treatment, respectively. As a result, both irrigation and N affect grain yield by affecting soil water status, improving above-ground biomass, and finally affecting yield components. In addition, I1800N60 also obtained a higher WUE and the highest NPFP due to a higher grain yield and a lower N application rate. Hence, our study recommends that irrigation with 1800 m^3^ ha^−1^ coupled with 60 kg N ha^−1^ could be a promising strategy for synergistically improving oilseed flax WUE, grain yield and yield components within this semi-arid region.

## 1. Introduction

Semiarid regions are characterized by the relatively low annual precipitation of 25 to 50 cm. For example, Cattivelli et al. [1] point out that 40% of the world’s cropland is arid or semi-arid and suffers from low annual rainfall, for example, in China. Climate change has changed the distribution of precipitation in time and space, which has resulted in large fluctuations in irrigation water demand in both annual and growth stages, especially in Northwest China [2]. In addition to drought, the soil fertility is seriously poor [3]. Fertilizing the soil with nitrogen becomes an important element of water and nitrogen management for expanding both crop growth and yield.

Oilseed flax (*Linum usitatisimum* L.), one of the most important oil crops, was harvested at 3.0 million tons worldwide and 0.33 million tons in China in 2019 [3]. The growth areas of oilseed flax in China can be divided into seven geographical areas with distinctive ecological characteristics [4]. Oilseed flax contributes significantly to society by providing food, livestock feed, industry, and raw materials, all of which are critical for economically stable agricultural production [5]. In addition, flaxseed is also used in the production of a vegetable oil known as flaxseed oil, which is edible and considered one of the oldest commercial oils [6]. The global production of flaxseed oil was 0.78 million tons, and China produced 0.23 million tons in 2019 [7]. The growing interest in flaxseed for food, feed, and industrial products means more attention needs to be focused on increasing its production to meet the growing demand [8]. Currently, increasing crop yields of oilseed flax in Northwest China for its economic value is considered an important research goal with a focus on managing nutrient availability and water content in permeable soils.

Water is the main factor restricting the effectiveness of nitrogen fertilizers under water-limiting conditions, and a sufficient water supply can enhance water and nutrient uptake [9]. Under low nitrogen conditions, the effect of water limitation is clear, and high levels of nitrogen application can make up for limitations in water availability and promote crop growth [10]. Wang et al. [11] stated that to facilitate nitrogen uptake, soil water availability in moist soil compartments should be maintained at a high level with a partial irrigation of the root zone. Despite this, high water levels cause serious environmental issues, such as greenhouse gas emissions and nitrogen leaching loss below the root zone and into groundwater [12], N leaching loss below the root zone and down into ground water groundwater pollution [13]. Therefore, irrigation and the application of nitrogen according to local conditions can produce a synergistic effect and can “promote fertilizer with water” and “adjust water with fertilizer”, which is of great significance for water conservation, fertilizer resources, and environmental protection.

Previous studies have reported that the grain yield is directly determined by EC (effective capsule, which flowers normally and produces a capsule; the branches hang capsule at maturity; ineffective capsule is unable to flower normally or shed the capsules after flowering; the branches are without capsules at maturity) number and grain number per capsule, and these yield components can be regulated by irrigation and nitrogen [14,15]. For example, Zhang et al. [16] reported that the N application rate significantly affected EC numbers and the grain number per capsule of oilseed flax. Zare et al. [17] reported that nitrogen stress occurring in the early season of oilseed flax may result in a loss in grain yield by reducing grain yield components. It can be seen that the influence of irrigation and nitrogen on oilseed flax grain yield is mainly through the influence of grain yield components. Clarifying the relationship between grain yield components, irrigation and nitrogen application amounts are, therefore, critical. There are two reasons that the challenge of water and nitrogen management establishes an effective ratio of water and nitrogen for optimizing soil nutrient availability, and gaining the expected grain yield in terms of desired quantity and quality. Firstly, while low N supply does not produce the expected outcome, soil water may be depleted under high N supply [18]. Secondly, crop response to nitrogen depends on soil water content; the amount and frequency of rainfall during crop-growing seasons may upset the desired soil water content during nitrogen applications [19]. Our study seeks to identify the optimal water-to-nitrogen ratio for irrigation and fertilization management, as well as to evaluate its effectiveness in terms of oilseed and flax grain yield, as well as water and nitrogen-use efficiency in a semiarid region of Northwest China.

## 2. Results

### 2.1. Weather Conditions

There was a large variation in rainfall between the two years, with more than half of the annual rainfall occurring from July to September (Figure 1A). The maximum monthly mean rainfall occurred in July 2017 and August 2018, respectively. Compared with the long-term average of 377 mm (1981–2010), the annual rainfall was higher in 2017 (407.4 mm) and 2018 (481.1 mm). Compared with the yearly average fallow rainfall (99.1 mm), the fallow rainfall was higher in 2017 (109.9 mm) and 2018 (120.2 mm). The seasonal rainfall in 2017 was similar to the yearly average, but it was higher in 2018 (360.9 mm). The annual rainfall during the wetter growing season (2018) was higher by 23% than the long-term average (1981–2010). Minimum temperatures occurred in January each year during the growing season, while maximum temperatures occurred at the young fruit stage in each growing season (Figure 1B).

### 2.2. Variance Analysis of GY, Water, and N Utilization-Related Traits

A combined ANOVA for GY, water, and N utilization-related traits revealed that there was a significant effect of irrigation, N, year, and their interactions on WUE (Table 1). The irrigation was significant for yield components (except thousand kernel weight) and water utilization. The nitrogen was significant for almost all GY-traits (except thousand kernel weight), water-use efficiency, and N utilization. The year was significant for GY, EC, GN, WUE, IWUE, and ANUE. A significant I × N and I × Y treatment interaction was detected for EC and WUE, and a significant I × N × Y interaction was observed for WUE (Table 1).

### 2.3. Water Storage Efficiency and Soil Water Storage

Due to irrigation occurring at the anthesis and kernel stages, the effects of irrigation on WSE and SWS before sowing in 2017 were not considered (Figure 2A,C and Figure 3A,B). Nitrogen application amounts had significant effects on WSE over the two growing seasons (*p* < 0.05). The WSE showed an increasing tendency with increasing nitrogen application amounts in 2017 (Figure 2A), while the WSE first decreased and then increased with increasing irrigation at the same N level in 2018 (Figure 2B). The WSE firstly increased and then decreased with increasing N at the I0 level, and increased with increasing N at the I1800 level. The WSE had an insignificant difference among I1200N0, I1200N60, and I1200N120 treatments in 2018 (Figure 3A). In the 2017 growing season, the WSE ranged from 31.85% to 40.41%, and the highest WSE was observed at N120; in the 2018 growing season, the WSE ranged from 29.84% to 52.29%, and the highest WSE was observed at I1800N120.

In the 2017 growing season, there was no significant difference in SWS among different N treatments before sowing (Figure 2C). Both nitrogen and irrigation had significant effects on SWS at harvest. In the 2017 growing season, the SWS after harvest first increased and then maintained stability with increasing irrigation amounts under the N120 level and increased with increasing nitrogen under the I1800 level (Figure 2E and Figure 3C). In the 2018 growing season, both nitrogen and irrigation had significant effects on SWS before sowing and after harvest. In the 2018 growing season, the SWS before sowing and after harvest increased with increasing irrigation under the same N level (Figure 2D,F).

The SWS before sowing increased with increasing irrigation under N120 level (Figure 3B), while the SWS at harvest decreased with increasing nitrogen under I1800 level (Figure 3D). In the 2017 growing season, the SWS at harvest ranged from 182.13 to 229.94 mm, and the highest SWS was observed at I1200N60. In the 2018 growing season, the SWS before sowing ranged from 219.54 to 281.20 mm, and the highest SWS was observed at I1800N120; the SWS at harvest ranged from 274.85 to 325.75 mm, and the highest SWS was observed at I1800N0.

### 2.4. Biomass, Grain Yield and Yield Components

The grain yield and biomass increased with the increasing irrigation application amount (0–1800 m^3^ ha^−1^) in 2017 and 2018 (Table 2). From I0 to I1800, the grain yield increased by 43.28% and 13.28% in the 2017 and 2018 growing seasons, respectively. However, the grain yield had an insignificant difference between the I1200 and I1800 levels in 2017 (*p* > 0.05). From I1200 to I1800, the grain yield only changed by 6.50% and 6.76% in both growing seasons. Nitrogen had a significant effects on grain yield in both growing seasons (*p* < 0.05). Compared with N0, the grain yield of N application treatments increased by 8.13%, 14.95% (2017) and 7.09%, 5.16% (2018), respectively. The biomass of N60 and N120 had insignificant differences over two growing seasons, but N120 was significantly higher than the N0 treatment (*p* < 0.05).

In 2017, the EC ranged from 143.47 to 340.04 × 10^5^ hm^−2^, and the highest EC was observed at I1800N60 (Table 3). The GN ranged from 6.73 to 8.03, and the highest GN was observed at I1200N60. In 2018, the EC ranged from146.13 to 366.19 × 10^4^ hm^−2^, and the highest EC was observed at I1800N60; the GN ranged from 6.53 to 7.53, and the highest GN was observed at I1200N120 (Table 3). The TKW ranged from 5.40 to 6.28 g in 2017, and from 7.43 to 8.00 g in 2018. In 2017, the EC increased with increasing nitrogen application amounts at I0 and I1200 levels, while the influence of the EC for nitrogen has no obvious law at I0 and I1200 levels in 2018. Over the two growing seasons, the EC first increased and then decreased with increasing nitrogen application amounts at the I1800 level (Figure 4A,B). The GN first decreased and then increased with increasing nitrogen application amounts at the I1200 and I1800 levels in 2017, but continued to increase at I1200 level and decrease at I1800 with increasing nitrogen application amounts in 2018. The GN first decreased and then increased with increasing nitrogen application amounts at the N0 level in two growing seasons (Figure 4C,D). At I0 and I1200 levels, the trend of TKW in 2017 and 2018 was opposite; however, the TKW continued to increase with increasing nitrogen application amounts at the I1800 level (Figure 4E,F). Irrigation significantly increased the EC and GN at the N60 level in 2017 (Figure 5A,C), but significantly decreased the EC and GN at the N120 level in 2018 (Figure 5B,D), and significantly increased the TKW at the N120 level in 2017 and at the N60 level in 2018 (Figure 5E,F). I1800N60 treatment achieved the highest GY, which was 1545.42 and 1281.67 kg ha^−1^ in 2017 and 2018, respectively. The GY of I1800N60 increased by 69.12% and 22.77% compared with I0N0 treatment in 2017 and 2018, respectively.

Over the two growing seasons, a positive relationship was observed between GY and EC (Figure 6A). There was a positive relationship between GY and GN, TKW in 2017 (Figure 6B,C), but a negative relationship existed between GY and GN, TKW in 2018 (Figure 6B,C), which indicates that the increased GY was due to the higher EC in this study.

### 2.5. Water Consumption, Water and Nitrogen-Use Efficiency

In two growing seasons, ET decreased with the increase in nitrogen under the same irrigation level and increased with the increase in the irrigation amount under the same nitrogen application rate (Table 4). The highest ET was observed at the I1800N0 treatment in 2017 and the I1800N120 treatment in 2018, which had significantly higher ET than other treatments. The highest WUE (3.55 kg ha^−1^ mm^−1^ and 2.03 kg ha^−1^ mm^−1^) of oilseed flax was observed at the I0N120 treatment in 2017 and at the I0N60 treatment in 2018.

The ANUE of oilseed flax showed decreasing trends with the increase in nitrogen application levels under the I1200 and I1800 levels, while it showed increasing trends with the increase in nitrogen application amounts under the I0 level. Compared with the I1200 treatment, the I1800 can significantly increase the ANUE under the N60 treatment but decrease it under the N120 treatment. The NPFP of oilseed flax showed decreasing trends with the increase in nitrogen application amounts under the same irrigation levels. Under the same nitrogen application conditions, irrigation can significantly increase the NPFP. The highest NPFP was observed at I1800N120 treatment over two growing seasons.

## 3. Discussion

The present study reports on the different changes in WSE and SWS of oilseed flax resulting from applications of irrigation combined with nitrogen under field conditions. The difference in irrigation and nitrogen levels significantly influenced the WSE and SWS. For both growing seasons, the application of N 120 kg ha^−1^ (N120) resulted in significantly higher water storage efficiency, relative to N0 and N60 under the I1800 treatment (Figure 2B and Figure 3A). Our results showed that the value of WSE was 29.84–52.29%. This value is in agreement with those of Shangguan et al. [20], who reported that the WSE of the Loess Plateau is about 30–35%. The best situation for WSE had occurred at I1800N120 of 52.29%, and the worst of it had occurred at I1200N0 of 29.84%. It is not surprising that irrigation sharply increased SWC, ultimately resulting in an increase in WSE [21]. The N benefits might be maximal under sufficient water conditions as a result of strong water × N interaction, which might be explained by the fact that the I1800N120 treatment had the highest WSE.

The SWS before sowing played an important role in improving the annual yield and water productivity of both wheat and maize [22], which is very important for germination and seedling establishment [23], and the seedling establishment phase is important for the productivity and profitability of crop production [24]. In our study, irrigation significantly increased the SWS before sowing in 2018 regardless of whether N was applied or not (Figure 2D). Meanwhile, N application significantly increased the SWS before sowing in 2018 under the I1800 level (Figure 3B). The SWS after harvest is different due to water–nitrogen coupling treatments, which are the result of the combined effects of precipitation, irrigation, crop production, and evapotranspiration. In our current study, irrigation significantly increased the SWS after harvest in 2017 and 2018 under the same N application rate (Figure 2E,F). The spatio-temporal dynamics of SWS revealed that the N application rates had a significant impact on SWS [25]. Our present study showed that N rate significantly improved the SWS before sowing in 2018 under the I1800 level; this may be related to the SWS after harvest in 2017. The N60 and N120 treatments significantly improved the SWS after harvest in 2017 under the I1800 level compared with N0. The results demonstrated that the irrigation water was not fully utilized and that part of it was stored in the soil for absorption and utilization by the next crop. The effects of high nitrogen (N120) on the SWS after harvest were different in 2017 and 2018 at I1800 level; N120 increased the SWS after harvest in 2017 and decreased the SWS after harvest in 2018. This is due to differences in precipitation during the growing season, which might be an indication that the major factor determining SWS is water supply (irrigation and precipitation) and that N has a minor effect on this parameter.

Generally, appropriate water and rational fertilization can increase crop yield [26]. The grain yield and biomass were increased in the current experiment by both irrigation and nitrogen application, but nitrogen had insignificant effects on grain yield in both growing seasons (Table 2). This result indicates that water was the main factor limiting oilseed flax grain yield, and nitrogen had a lower effect on oilseed flax grain yield than water. There is a close relationship between GY and water consumption, and GY is largely determined by evapotranspiration [25]. In this study, irrigation significantly increased the ET, but N had insignificant effects on ET in both growing seasons under the same irrigation level, partially explaining the similar GY among the three N application rates. Our findings showed that irrigation contributed more to GY, while N increased GY, such N benefits may be minimal under adequate water conditions due to strong water N interaction.

Crop yield components play a vital role in determining grain yield and these can be affected by variety, environmental conditions, and agronomic management [27]. Previous studies have demonstrated that crop yield components can be regulated by irrigation and nitrogen [28,29]. In this study, increasing irrigation under the same nitrogen level significantly increased the EC, as did increasing nitrogen application amounts under the same irrigation level (Table 3). The GY increased with the increase of EC, which is in line with the previous result that EC had positive effects on GY [30]. Grain yield was positively correlated with GN and TKW in 2017, but negatively correlated with GN and TKW in 2018 (Figure 6B,C). The different results between 2017 and 2018 could be explained by the competition among yield components of oilseed flax. Although the application of nitrogen had a negative impact on TKW under the I1800 level (Figure 4E,F), these losses can be compensated by increased EC, which finally achieved a higher grain yield (Table 2 and Table 3). I1800N60 treatment achieved the highest grain yield (Table 3). These results indicate that irrigation of 1800 m^3^ ha^−1^ with 60 kg N ha^−1^ could meet the demand of oilseed flax for water and nitrogen in the test area and in the area with the same ecological conditions. This is consistent with the result of our previous study [28].

The interaction of irrigation levels and nitrogen application had a significant effect on WUE and WUEI (Table 1). Although improving water and fertilizer resource utilization efficiency is particularly important for sustainable agricultural development in the context of water shortages and over-fertilization, high-yield and high-resource use are not compatible [26]. Usually, high WUE and nitrogen-use efficiency are obtained under deficit irrigation and relatively low N application levels, but this will always be accompanied by relatively low yields. When high yields are obtained, WUE and nitrogen-use efficiency will be lower. In this study, the highest WUE (3.55 kg ha^−1^ mm^−1^ in 2017, 3.72 kg ha^−1^ mm^−1^ in 2018) of oilseed flax was observed at the I0N120 and I0N60 treatments (Table 4). Irrigation and nitrogen application amounts significantly improved WUEI; a similar result was also reported by Zotarelli et al. [31], who reported that water–nitrogen coupling experiments in tomato revealed that WUEI was significantly affected by the interaction between water and nitrogen. The WUEI was aided by nitrogen supply, and this improvement was more pronounced at lower irrigation levels [32]. Our results also showed that the WUEI of I1200 was 38.34% and 36.38% higher than that of I1800 under N60 treatment and 43.38% and 43.86% higher than I1800 under N120 treatment, both in two growing seasons.

In the present study, the ANUE increased first and then decreased under the irrigation levels; a similar result was demonstrated by Wu et al. [33], who reported that excessive nitrogen application decreased the ANUE. NPFP is considered one of the most important indicators for cereal crops, directly reflects N input into crop production systems and its economic return [34]. Irrigation and N application plays an important role in improving NPFP, but excessive N fertilizer has a negative effect on NFP [2,23]. In our study, irrigation significantly increased the NPFP of oilseed flax under the same nitrogen application level but it decreased with nitrogen application level increase. A similar result was also reported by Ma et al. [35] and Quan et al. [36], who reported that the excessive N led to reduced NFP.

## 4. Materials and Methods

### 4.1. Experimental Site

The field experiment was conducted in Gansu Province, China, by the Dingxi Academy of Agricultural Science (34.26° N, 103.52° E, altitude 2050 m) over two seasons, 2017 and 2018, and lasted from April to August in each season. The soil type is dark loessial soil (Heilutu), with 50% sand, 20% silt, and 30% clay. For 2017 and 2018, the annual climatic data collected in the experimental field station were: rainfall with a mean of 405.9 mm and a mean temperature of 8.8 °C with an average of 2161 h of sunshine duration. In 2017, oilseed flax was planted on April 7th and harvested on August 8th, but in 2018 the planting and harvesting were on April 5th and August 12th, respectively. The growth of the oilseed flax was categorized into five stages from sowing seeds to harvesting: seedling, budding, anthesis, kernel, and maturity stages. Before planting oilseed flax, several soil samples were collected from 30 cm below the surface (0–30 cm) to determine the basic nutrient content as the baseline for assigning the amount and frequency of N application used for the research [37]. These soil samples were randomly collected from five sites in the experimental field before the study. The physical and chemical characteristics of the 0–30 cm soil layer are provided at Table 5.

### 4.2. Experimental Design

The split-plot design was used in the field experiment, with irrigation as the main or whole-plot factor and nitrogen as a sub-factor or split-plot factor; 9 combinations of treatments were included in the experiment and with three replications. The main plot was 6.8 m × 5.0 m, while the sub-plot was 5.0 m × 2.0 m. They involved the applications of the amounts of water supply in meters, and nitrogen in kilograms as shown in Table 6. For irrigation, water was supplied by surface (flood) irrigation with a pipe attached to a flow meter for measuring the amount of water applied [38]. In order to ensure the uniform irrigation of the plots, 40 cm-high ridges were set up in each plot to make the ground as flat as possible. I1200 supplied 1200 m^3^ ha^−1^, i.e., 600 m^3^ ha^−1^ for anthesis and the kernel stages, respectively. Similarly, I1800 supplied 900 m^3^ ha^−1^ for each stage of anthesis and the kernel stage. N60 application, 60 kg N ha^−1^, 40 kg N ha^−1^ and 20 kg N ha^−1^ were applied to the soil before sowing and at the budding stage. For the N120 application, 120 kg N ha^−1^, 80 kg N ha^−1^ and 40 kg N ha^−1^ were applied to the soil before sowing and at the budding stage. N fertilizer is urea (N content 46%); an additional 90 kg P_2_O_5_ ha^−1^ was applied to every plot before sowing; phosphate fertilizer is calcium superphosphate (P_2_O_5_ 16%).

### 4.3. Measurement and Calculation

The measurement of the treatment results aimed to identify an irrigation and nitrogen ratio for the management of water and nitrogen supplies. Grain yield and yield components, PSE, SWS, water consumption (ET), and water and nitrogen-use efficiency were analyzed to investigate the impacts of water and nitrogen coupling management during the oilseed-flax-growing season.

The following equations were used to calculate water and N use efficiency [11,39,40,41]:WSE = ΔSWS_f_/R_f_ × 100(1)
ΔSWS_f_ = SWS_ph_ − SWS_b_(2)
where WSE (%) is water storage efficiency, ΔSWS_f_ (mm) is soil water storage during the fallow period, R_f_ (mm) is the rainfall during the fallow period, SWS_ph_ (mm) is soil water storage at the previous harvest or the beginning of the fallow period, and SWS_b_ (mm) is the soil water storage at oilseed flax planting or the end of the fallow period.
SWS = ∑SWC_i_·D_i_·H_i_(3)
where SWS (mm) is soil water storage in the 0–160 cm, and i = 1, 2, 3... 8, SWC (%) is the soil water content, D_i_ (g cm^−3^) is the soil bulk density, H_i_ (cm) is the soil depth.
SWC = (FW − DW)/(DW − AW) × 100(4)
where FW (g) is the fresh weight of the soil sample with the aluminum box, DW (g) is the dry weight of the soil sample with the aluminum box, and AW (g) is the weight of the aluminum box. The soil water content (SWC) in the 0–160 cm layer was measured every 20 cm before sowing and after harvesting, and 5 points were randomly selected in each plot.
ET= SWS_bf_ − SWS_h_ + P_i_(5)
where ET (mm) is water consumption, SWS_bf_ (mm) is soil water storage before the sowing stage, SWS_h_ (mm) is soil water storage after harvesting, and Pi (mm) is precipitation during the crop-growing period.
WUE = GY/ET(6)
WUEI = GY/I(7)
where WUE (kg ha^−1^ mm^−1^) is water-use efficiency, GY (kg ha^−1^) is grain yield, WUEI (kg ha^−1^ mm^−1^) is irrigation water-use efficiency, and I (m^3^ ha^−1^ was convert to mm) is irrigation amount. At the end of the growing season, the plants in each plot were hand-harvested for grain yield.
ANUE = (GY at Nx − GY of N0)/N rate(8)
NPFP = GY/N rate(9)
where ANUE (kg kg^−1^) is agronomy N use efficiency, Nx is N treatment, N0 is no N application, NPFP (kg kg^−1^) is N partial factor productivity.

### 4.4. Statistical Analyses

The value of each indicator was the mean of three replicates per treatment, and the SPSS statistical package v.24.0 (SPSS Inst., Cary, NC, USA) was used to perform the analysis of variance. Data pre-processing was performed using Excel 2016. The figures were plotted using Origin 2019b (Systat Software Inc., San Jose, CA, USA). All pair-wise comparisons of the treatment means were performed using the least significant difference (LSD) test with significance determined at the 5% level.

## 5. Conclusions

In this region, irrigation (1800 m^3^ ha^−1^) coupled with 60 kg N ha^−1^ (I1800N60) obtained the highest effective capsule, grain yield and N partial factor productivity. I1800N60 treatment increased yield by more than 69.04% and 22.80%, respectively, when compared with I0N0 treatment in 2017 and 2018. The positive relationship between grain yield, and EC number was observed, and increased EC number contributed to I1800N60 treatment’s increased yield. Water deficit was the main factor effecting the SWS of oilseed flax in the semi-arid area of Northwest China. The interaction of irrigation levels and nitrogen application increases the water storage efficiency, which is beneficial for the next crop-growing season. Thus, selecting irrigation of 1800 m^3^ ha^−1^ combined with nitrogen of 60 kg ha^−1^ could be a better field management option for oilseed flax fields in the semi-arid areas of China.

## Figures and Tables

**Figure 1 plants-12-00051-f001:**
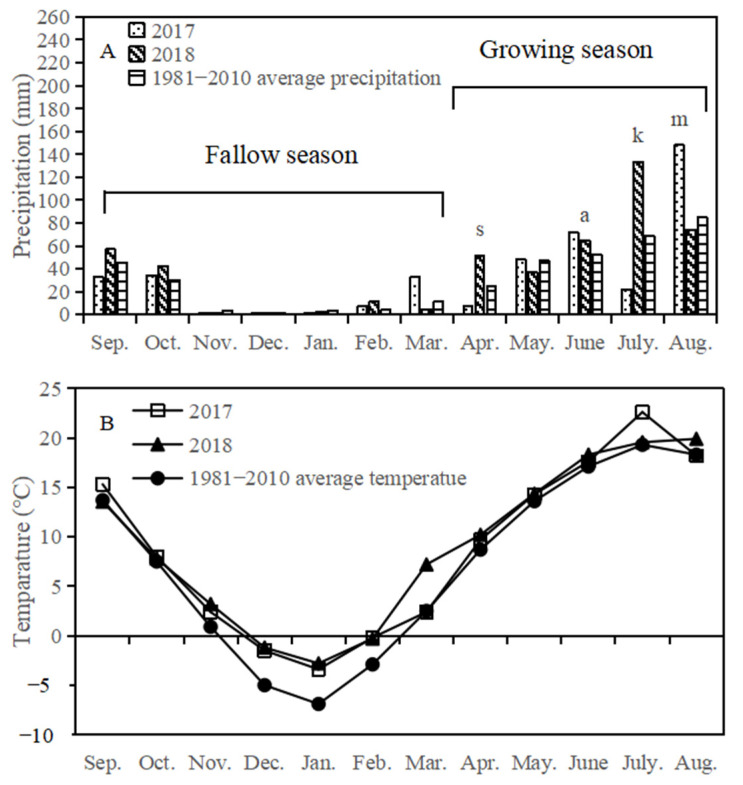
Monthly precipitation (**A**) and temperature (**B**) in 2017, 2018 and 1981–2010 at Dingxi station, China. The letters indicate sowing (**s**), anthesis (**a**), kernel (**k**), and maturity (**m**). The growth of the oilseed flax was divided into five stages: seedling, budding, anthesis, kernel and maturity stages.

**Figure 2 plants-12-00051-f002:**
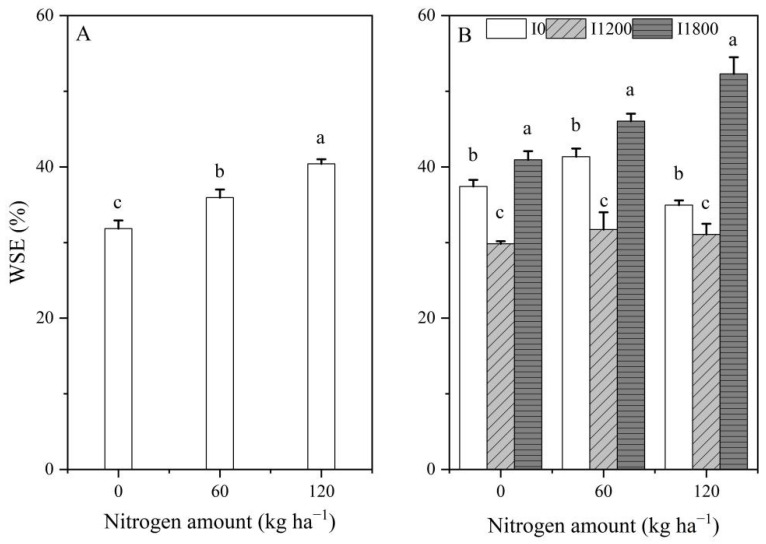
Response of WSE to nitrogen application rate in the 2017 (**A**) and 2018 (**B**); Response of before-sowing SWS to the N application rate in 2017 (**C**) and 2018 (**D**); Response of after-harvest SWS to the N application rate in 2017 (**E**) and 2018 (**F**). Data are averaged across treatments. Error bars show the standard error of the mean. WSE is water storage efficiency (%), SWS is soil water storage (mm). Different lowercase letters represent the least significant difference (LSD) at *p* < 0.05.

**Figure 3 plants-12-00051-f003:**
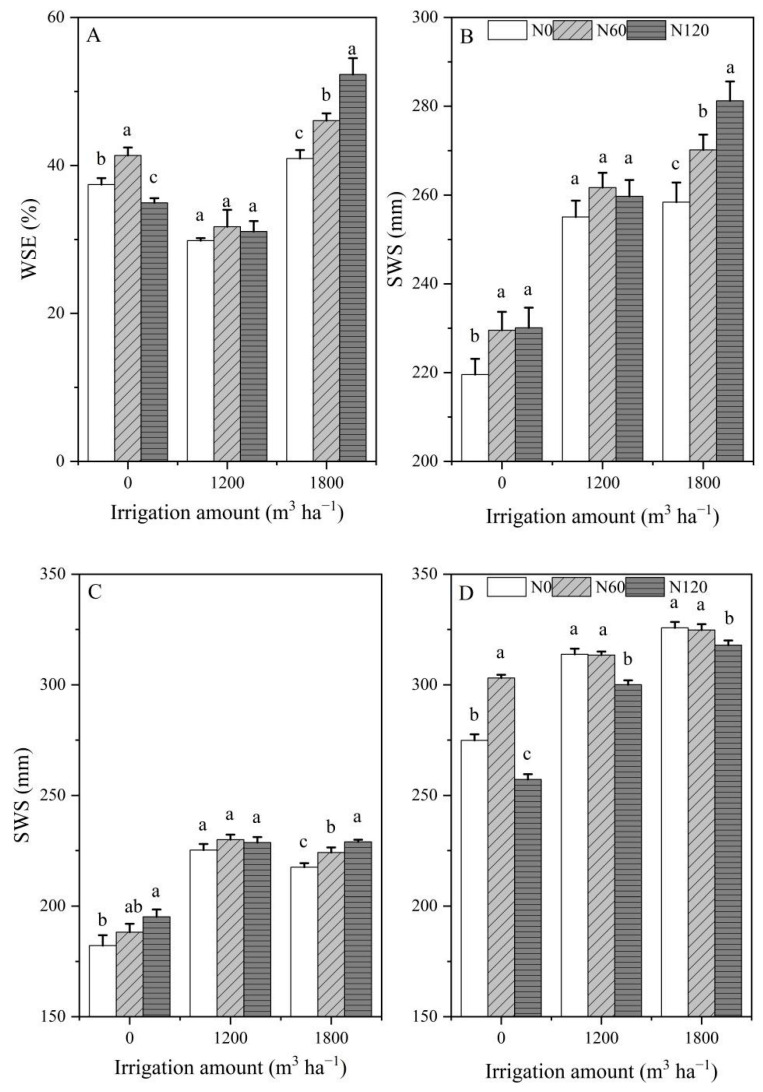
Response of WSE (**A**) and before sowing SWS (**B**) to irrigation amount in 2018; Response of after-harvest SWS to irrigation amount in 2017 (**C**) 2018 (**D**). Data are averaged across treatments. Error bars show the standard error of the mean. WSE is water storage efficiency (%), SWS is soil water storage (mm). Different lowercase letters represent the least significant difference (LSD) at *p* < 0.05.

**Figure 4 plants-12-00051-f004:**
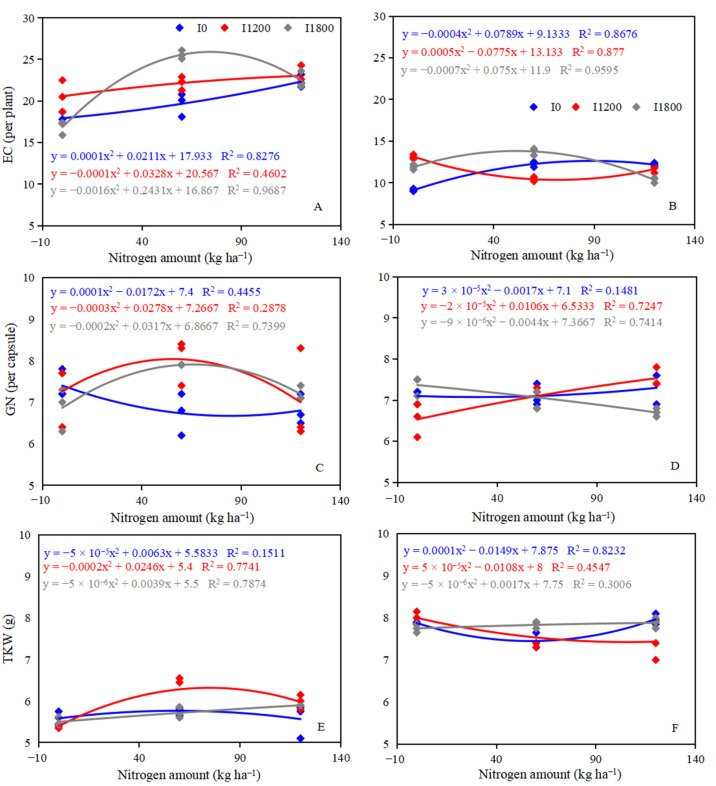
The yield component response to the N application rate in 2017 (**A**,**C**,**E**) and 2018 (**B**,**D**,**F**). EC, effective capsule number; GN, grain number; TKW, thousand kernel weight.

**Figure 5 plants-12-00051-f005:**
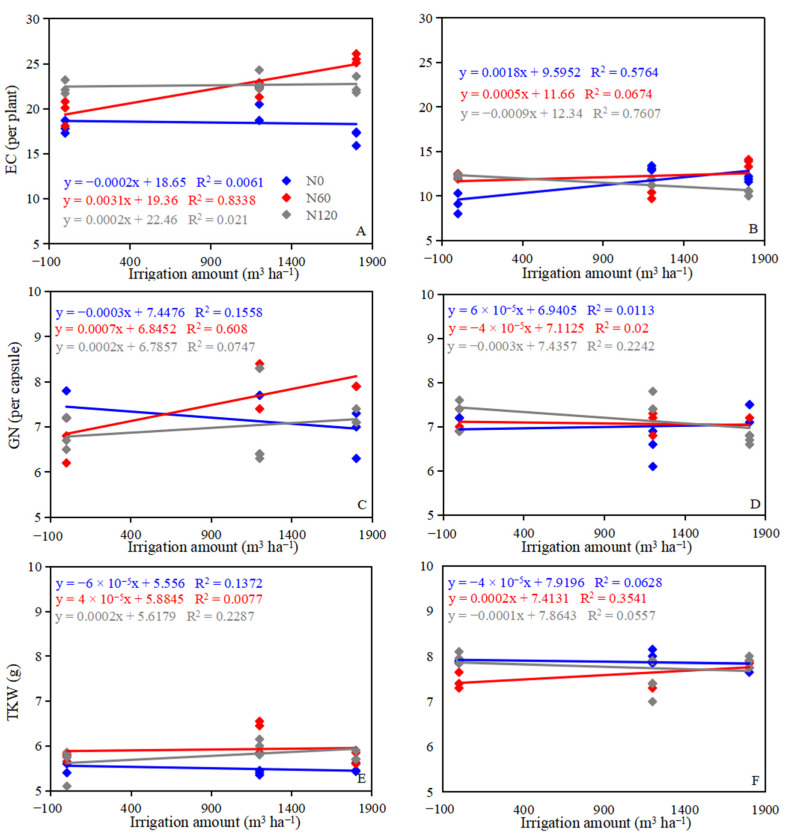
The yield component response to irrigation amount in the 2017 (**A**,**C**,**E**) and 2018 (**B**,**D**,**F**). A and B, EC, effective capsule number; GN, grain number; TKW, thousand kernel weight.

**Figure 6 plants-12-00051-f006:**
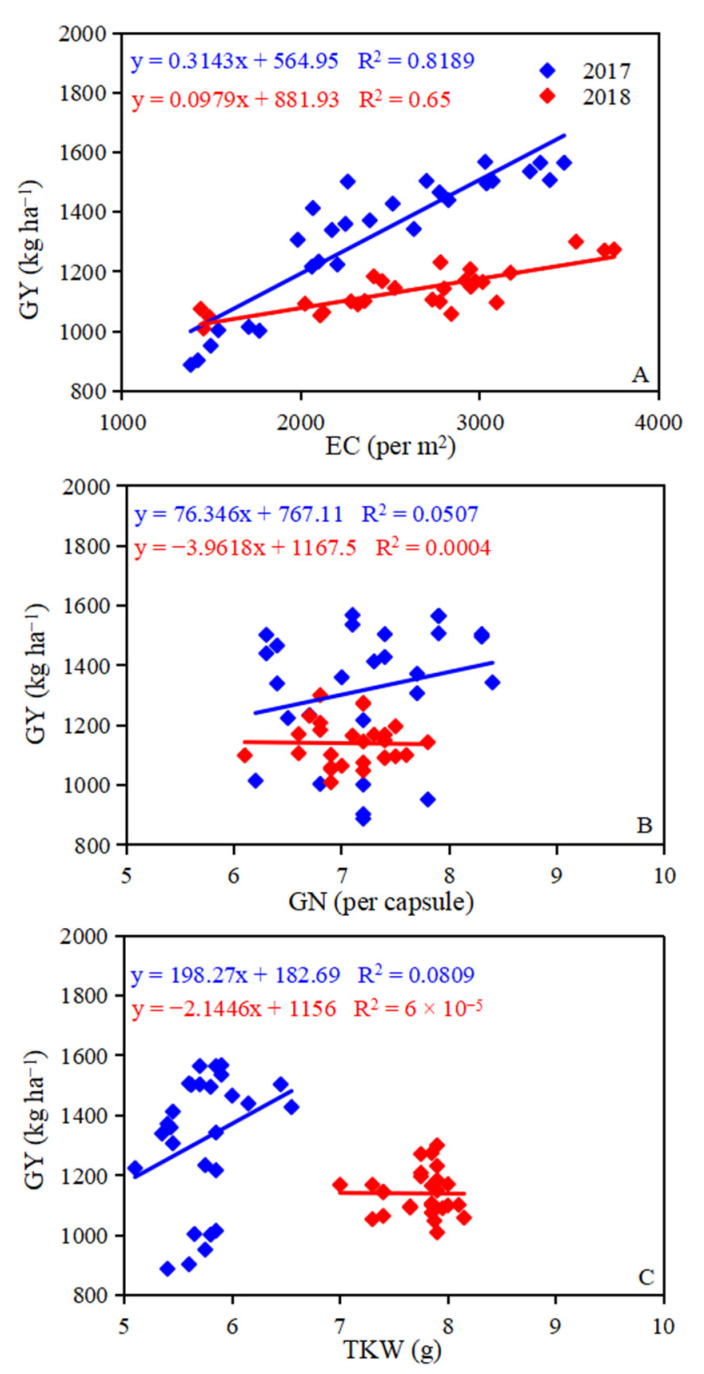
The relationships between GY and EC (**A**), GN (**B**), and TKW (**C**) in the 2017 and 2018. GY is grain yield, EC is effective capsule, GN is grain number, and TKW is thousand kernel weight.

**Table 1 plants-12-00051-t001:** Combined analysis of variance for traits related to yield components, water, and N utilization.

		Irrigation (I)	Nitrogen (N)	Year	I × N	I × Y	N × Y	I × N× Y
	df	2	2	1	4	2	2	4
Yield components	Grain yield (kg ha^−1^)	***	**	*	*	ns	ns	ns
	Biomass (kg ha^−1^)	*	**	ns	ns	ns	ns	ns
	Effective capsule number (per plant)	*	*	*	**	*	ns	ns
	Grain number (per capsule)	**	*	*	*	ns	ns	ns
	Thousand kernel weight (g)	ns	ns	ns	ns	ns	ns	ns
Water utilization	Water consumption (mm)	**	ns	ns	*	ns	ns	ns
	Water-use efficiency (kg ha^−1^ mm^−1^)	**	**	**	**	**	**	**
	Irrigation water-use efficiency (kg ha^−1^ mm^−1^)	**	ns	*	*	ns	ns	ns
N utilization	Agronomy N use efficiency (%)	ns	**	*	*	ns	*	ns
	N partial factor productivity (kg hg^−1^)	ns	**	ns	*	ns	*	ns

ns denotes *p* > 0.05; *, **, and *** represent significance at the 0.05, 0.01 and 0.001 probability levels, respectively.

**Table 2 plants-12-00051-t002:** Effects of different irrigation and nitrogen application on grain yield and biomass of oilseed flax.

Treatments		2017				2018			
		GY (kg ha^−1^) *	Changes (%)	Bio (kg ha^−1^)	Changes (%)	GY (kg ha^−1^)	Changes (%)	Bio (kg ha^−1^)	Changes (%)
Irrigation	I0	1048.30 b	–	2336.99 b,**	–	1070.22 c	–	2634.67 b	–
	I1200	1410.32 a	34.53%	2776.85 a	18.82%	1135.56 b	6.11%	2931.69 ab	11.27%
	I1800	1501.97 a	6.50%	2920.72 a	5.18%	1212.33 a	6.76%	3121.95 a	6.49%
Increase in percentage from I0 to I1800		43.28%		24.98%		13.28%		18.49%	
Nitrogen	N0	1225.89 b	–	2198.81 b	–	1094.67 b	–	2487.59 c	–
	N60	1325.55 ab	8.13%	3032.67 a	37.92%	1172.33 a	7.09%	3275.31 a	31.67%
	N120	1409.15 a	6.31%	2803.09 a	−7.57%	1151.11 a	−1.81%	2925.41 b	−10.68%
Increase in percentage from N0 to N120		14.95%		27.48%		5.16%		17.60%	

* GY, grain yield; Bio, biomass. ** Different lowercase letters represent least significant difference (LSD) at *p* < 0.05.

**Table 3 plants-12-00051-t003:** Response of yield components to different irrigation and nitrogen treatments in the 2017 and 2018 growing seasons.

Treatments	2017				2018			
	EC (×10^5^ ha^−1^) *	GN (No.)	TKW (g)	GY (kg ha^−1^)	EC (×10^5^ ha^−1^)	GN (No.)	TKW (g)	GY (kg ha^−1^)
I0N0	143.47 g,**	7.40 b	5.58 bcd	913.79 g	146.13 g	7.10 ab	7.88 ab	1044.00 e
I0N60	167.17 f	6.73 c	5.77 bcd	1006.50 f	208.53 f	7.10 ab	7.45 c	1069.67 de
I0N120	212.17 e	6.80 c	5.57 cd	1224.61 e	231.80 e	7.30 a	7.97 a	1097.00 d
I1200N0	218.01 e	7.27 bc	5.40 d	1339.17 d	278.43 c	6.53 c	8.00 a	1087.67 de
I1200N60	261.57 d	8.03 a	6.28 a	1424.71 c	246.23 d	7.10 ab	7.53 bc	1165.67 bc
I1200N120	287.92 c	7.00 bc	5.98 ab	1467.07 bc	290.c	7.53 a	7.43 c	1153.33 c
I1800N0	219.27 e	6.87 c	5.50 cd	1424.71 c	309.40 b	7.37 a	7.75 abc	1152.33 c
I1800N60	340.04 a	7.90 a	5.72 bcd	1545.42 a	366.19 a	7.07 ab	7.83 abc	1281.67 a
I1800N120	312.75 b	7.20 bc	5.83 bc	1535.77 ab	288.19 c	6.70 bc	7.88 ab	1203.00 b

* EC, effective capsule number per plant; GN, grain number per capsule; TKW, thousand kernel weight. ** Different lowercase letters represent least significant difference (LSD) at *p* < 0.05.

**Table 4 plants-12-00051-t004:** Effects of different nitrogen and irrigation treatments on evapotranspiration, water-use efficiency, and nitrogen-use efficiency of oilseed flax in the 2017 and 2018 growing seasons.

Treatments	2017					2018				
	ET *	WUE	WUEI	ANUE	NPFP	ET	WUE	IWUE	NAUE	NPFP
	(mm)	(kg ha^−1^ mm^−1^)	(kg ha^−1^ mm^−1^)	(kg ha^−1^)	(kg ha^−1^)	(mm)	(kg ha^−1^ mm^−1^)	(kg ha^−1^ mm^−1^)	(kg ha^−1^)	(kg ha^−1^)
I0N0	358.12 d,**	2.55 e	–	–	–	305.59 g	3.42 b	–	–	–
I0N60	352.05 d	2.86 d	–	1.55 c	16.78 c	287.39 h	3.72 a	–	0.43 c	17.83 c
I0N120	345.12 e	3.55 a	–	2.59 a	10.21 e	333.79 f	3.29 c	–	0.44 c	9.14 e
I1200N0	435.03 c	3.08 c	11.16 b	–	–	422.18 e	2.58 e	9.07 b	–	–
I1200N60	430.31 c	3.31 b	11.87 a	1.43 cd	23.75 b	429.12 e	2.71 d	9.71 a	1.30 b	19.43 b
I1200N120	431.64 c	3.40 ab	12.23 a	1.07 de	12.23 d	440.54 d	2.62 de	9.61 a	0.55 c	9.61 d
I1800N0	502.79 a	2.83 d	7.91 c	–	–	473.53 c	2.45 f	6.40 d	–	–
I1800N60	496.14 b	3.11 c	8.58 c	2.01 b	25.76 a	486.30 b	2.64 de	7.12 c	2.16 a	21.36 a
I1800N120	491.34 b	3.12 c	8.53 c	0.93 e	12.80 d	504.21 a	2.38 f	6.68 d	0.42 c	10.03 d

* ET, water consumption; WUE, water-use efficiency; WUEI, irrigation water-use efficiency; ANUE, agronomy N use efficiency; NPFP, N partial factor productivity. ** Different lowercase letters represent least significant difference (LSD) at *p* < 0.05.

**Table 5 plants-12-00051-t005:** Soil porosity and chemical characteristics of the soil of the experimental site.

Year	Organic Matter (g kg^−1^)	Total N (g N kg^−1^)	Total P (mg P kg^−1^)	Alkali-Hydrolyzable N (mg N kg^−1^)	Available P (mg P kg^−1^)	Available K (mg K kg^−1^)	pH
2017	10.51	1.00	0.85	47.91	26.43	108.3	8.1
2018	10.12	0.81	0.69	49.37	27.24	108.42	8.1

**Table 6 plants-12-00051-t006:** Split-plot design.

Sub-Factor: Nitrogen (N)	Main Factor: Irrigation (I)			
	N|I	I0 (0 m^3^ ha^−1^)	I1200 (1200 m^3^ ha^−1^)	I1800 (1800 m^3^ ha^−1^)
	N0 (0 kg ha^−1^)	I0N0	I1200N0	I1800N0
	N60 (60 kg ha^−1^)	I0N60	I1200N60	I1800N60
	N120 (120 kg ha^−1^)	I0N120	I1200N120	I1800N120

## Data Availability

All data included in this study are available upon request by contact with the first author.

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
