# Peer review of "Water and Nitrogen Coupling Increased the Water-Nitrogen Use Efficiency of Oilseed Flax"

_plants, 2022, doi:10.3390/plants12010051_

Round 1
Reviewer 1 Report
General
The article presents the results of a field experiment, in which the combined effect of irrigation and N fertilization on flax yields was tested in north-west China. The results are interesting, although not very innovative. The major problem of the article is that it uses a very elaborate and complex approach to present a simple issue. Specific comments will follow below.
Language
The article requires thorough language editing.
Units
The use of units throughout the article is inconsistent e.g. for amounts of water applied you sometimes use mm, sometimes m3 ha-1 and sometimes m3 hm-1. The use of units must be consistent.
Introduction
The introduction is too long. Sometimes it provides information which is common knowledge. On the other hand, there is no information about the extension of this crop in China.
Line 40 – You refer to flax as an oil crop, but there is no reference in the data to the oil yield, only to the grain yield and the total above-ground plant production.
Line 65 – The major threat of over irrigation combined with N fertilization is N loss below the root zone and down into ground water.
EC number – If you refer to effective capsule number, are there ineffective capsules? How do you differentiate between the two? What is the definition of an effective capsule? Sometimes you use the term "capsule" sometimes "pod" and sometimes "kernel"…..
Lines 99-102 – This is a repetition of what had been said before.
Materials and Methods
Lines 105-122 – All this paragraph does not belong here. It belongs in the Introduction section. It also repeats some statements from the Introduction.
Line 126 – This is misleading the reader. The experiment was carried out during two seasons: 2017 and 2018 and lasted from April to August in each season.
Lines 137-139 – This information will be presented better in a Table.
Figure 1 – Should be presented in the Results section.
Line 147 – It should be 9 plots, no? What was the size of each plot/treatment?
Lines 156-158 – Water was applied by surface (flood) irrigation, I guess. Please, specify it. The irrigation methods determine, to a large extent, N availability and uptake. Are you sure water distribution over the surface was uniform?
Lines 156-158 – Please, provide a Table or a Figure where both irrigation and rain events along the growing season of flax in the experiment are indicated.
Lines 162-164 – 2/3 of the N was applied before sowing. This means, that a certain amount of the N could have been leached below the root zone. What was the soil type in the experimental site? Please, give soil texture data.
Lines 169-173 – This paragraph is not necessary. The readers know what a split-plot design is. Give them some credit….
Lines 181-190 – You determined soil water content gravimetrically (eq. 4). To what depth did you take soil samples?
Lines 193-195 – Eta is usually used for indicating evapotranspiration, measured by a class A pan or calculated using the Penman or Penman-Monteith equation. The use of Eta in this case is confusing. I suggest using another term.
Eq. 9 – Similarly to equations 6, 7 and 8, eq. 9 calculates NUE.
Please, describe how yield and yield components were assessed.
Results
Table 2 – In eq. 9 you define NFP (N fertilizer productivity) as GY/N application rate and the units are kg kg-1. In Table 2 N fertilizer productivity units are kg ha-1.
Line 283 – In this line you indicate that nitrogen had insignificant effects on grain yield in both growing seasons. However, in the statistical analysis results (Table 2), you indicate that N had a very significant effect on grain yield. How do you explain this contradiction?
Lines 284-286 – your results (Table 3) do not support this statement. On the contrary. Your results give an indication (not significant though) that over-fertilization impairs production.
Tables 2 and 3 – The results of the statistical analysis in Table 2 are not always in agreement with the results presented in Table 3. Please, check your statistical analysis.
Figure 6 – Change Y axis from kg hn-2 to kg hm-2
Table 5 – Very unclearly presented. Not possible to read it.
Discussion
In general, the discussion is not clearly organized. There are many repetitions, many sections belong to the conclusion section and I suggest to re-write and re-organize the discussion and make it more concise and to the point.
Lines 348-350 – This statement is not really correct. Your results are in agreement with those of Shangguan et al.
Lines 351-352 and Eq.1 – According to eq.1, PSE is affected only by natural precipitation (rainfall), assuming that irrigation does nor occur during the fallow period. In lines 351-352 you state that PSE can be affected by irrigation. Irrigation can affect the PSE value of the subsequent season (residual water) but in this case, the term PSE is not correct and I suggest to use the term WSE (Water storage efficiency), which refers to both precipitation and irrigation.
Lines 355-358 – How would N level increase PSE? Under elevated N levels, biomass production increases and therefore also plant water consumption, which leaves less water in the soil and leads to a lower PSE and SWS.
Lines 367-379 – This paragraph describes, again, the fact that under high N application levels SWS was higher. Please, provide an explanation to this strange phenomenon. There is no consistency between the two years, apparently because of differences in precipitation, which might be an indication that the major factor determining SWS is water supply (irrigation + precipitation) and that N has a minor effect on this parameter.
Lines 453-463 – In a previous paragraph, emphasis was given to WUE and NUE. In this paragraph, the emphasis in on grain yield. Usually, high WUE and NUE values will be obtained under deficit irrigation and relatively low N application levels, but this will always be accompanied by relatively low yields. When high yields are obtained, WUE and NUE will be lower and the optimum point is a matter of economic considerations. At the bottom line, the producer does not sell WUE or NUE but kilograms of product and therefore he would want to maximize his income at a minimum cost.

Author Response
Dear Reviewers:
Thank you for your letter and for the reviewers’ comments concerning our manuscript entitled “Water and nitrogen coupling increased the water-nitrogen use efficiency of oilseed flax” (ID: plants-2051341). Those comments are all valuable and very helpful for revising and improving our paper, as well as the important guiding significance to our researches. We have studied comments carefully and have made correction which we hope meet with approval. Revised portion are marked in red in the paper. The main corrections in the paper and the responds to the reviewer’s comments are as flowing:
- Language: The article requires thorough language editing.
Response: We invited our colleague (Zechariah Effah), who at CSIR-Plant Genetic Resources Research Institute now, helped to edit the language. He is very experienced in this field.
- Units: The use of units throughout the article is inconsistent e.g. for amounts of water applied you sometimes use mm, sometimes m3 ha-1 and sometimes m3 hm-1. The use of units must be consistent.
Response: We unified the units in the manuscript, especially the irrigation amount, as m3 ha-1, and all “hectare” units as “ha”. Such as L21, L27, L28, L240, L402, L411, and so on. In addition to this, the units in Table 3, Table 4, Table 5, Figure 2, Figure 3, Figure 4, Figure 5, Figure 6 were unified.
- Introduction: The introduction is too long. Sometimes it provides information which is common knowledge. On the other hand, there is no information about the extension of this crop in China.
Response: The author has condensed the abstract and added some content, and the specific modification as follows:
L42-45, Oilseed flax (Linum usitatisimum L.), one of the most important oil crops, which was harvested on 3.0 million tons worldwide and 0.33 million tons in China in 2019 [3]. The growth areas of oilseed flax in China can be divided into seven geographical areas with distinctive ecological characteristics [4].
Line 40– You refer to flax as an oil crop, but there is no reference in the data to the oil yield, only to the grain yield and the total above-ground plant production
Response: L47-50, In addition, flaxseed is also produced into vegetable oils known as flaxseed oil, which is edible and considered one of the oldest commercial oils [6]. The global production of flaxseed oil was 0.78 million tons and China produced 0.23 million tons in 2019 [7].
Line 65 – The major threat of over irrigation combined with N fertilization is N loss below the root zone and down into ground water
Response: L60-65, Wang et al. [11] stated that to facilitate nitrogen uptake, soil water availability in moist soil compartments should be maintained at a high level with partial irrigation of the root zone. Despite this, high water levels cause serious environmental issues such as greenhouse gas emissions and nitrogen leaching loss below the root zone and into groundwater [12], N leaching loss below the root zone and down into ground water groundwater pollution [13].
EC number – If you refer to effective capsule number, are there ineffective capsules? How do you differentiate between the two? What is the definition of an effective capsule? Sometimes you use the term "capsule" sometimes "pod" and sometimes "kernel"
Response: L70-72, (effective capsule: it flowers normally and produces a capsule; the branches hang capsule at maturity; ineffective capsule: it is unable to flower normally or shed the capsules after flowering; the branches are without capsules at maturity.)
Has been unified in manuscript as “capsule”. Such as L73, L75, Table 3, Figure 4, Figure 5.
Lines 99-102 – This is a repetition of what had been said before.
Response: The author deleted the duplicate content.
- Materials and Methods
Lines 105-122 – All this paragraph does not belong here. It belongs in the Introduction section. It also repeats some statements from the Introduction
Response: The author deleted the duplicate content, and added part of the content in the Introduction.
Line 126 – This is misleading the reader. The experiment was carried out during two seasons: 2017 and 2018 and lasted from April to August in each season
Response: L93-96, The field experiment was conducted in Gansu Province, China, by the Dingxi Academy of Agricultural Science (34.26°N, 103.52°E, altitude 2050 m) over two seasons: 2017 and 2018 and lasted from April to August in each season.
Lines 137-139 – This information will be presented better in a Table.
Response: This part of the content is replaced by a table.
Table 1. Soil porosity and chemical characteristics of the soil of the experimental site.
|
Year |
Organic matter (g kg−1) |
Total N (g N kg−1) |
Total P (mg P kg−1) |
Alkali-hydrolyzable N (mg N kg−1) |
Available P (mg P kg−1) |
Available K (mg K kg−1) |
pH |
|
2017 |
10.51 |
1.00 |
0.85 |
47.91 |
26.43 |
108.3 |
8.1 |
|
2018 |
10.12 |
0.81 |
0.69 |
49.37 |
27.24 |
108.42 |
8.1 |
Figure 1 – Should be presented in the Results section.
Response: Figure 1 had been presented in the Results section.
Line 147 – It should be 9 plots, no? What was the size of each plot/treatment?
Response: L111-113, 9 combinations of treatments were included in the experiment and with three replications. The main plot was 6.8 m × 5.0 m while the sub-plot was 5.0 m × 2.0 m.
Lines 156-158 – Water was applied by surface (flood) irrigation, I guess. Please, specify it. The irrigation methods determine, to a large extent, N availability and uptake. Are you sure water distribution over the surface was uniform?
Response: L115-118, For irrigation, water was supplied by surface (flood) irrigation with a pipe attached to a flow meter for measuring the amount of water applied [21]. In order to ensure the uniform irrigation of the plots, each plot set up 40 cm−high ridges to make the ground as flat as possible.
Lines 156-158 – Please, provide a Table or a Figure where both irrigation and rain events along the growing season of flax in the experiment are indicated.
Response: Figure 1 includes the rainfall during flax flax growth period, and flax irrigation occurs at anthesis and the kernel stage respectively, both of which are irrigated in the same amount.
L118-120, I1200 supplied 1200 m3 ha−1, i.e., 600 m3 ha−1 for anthesis and the kernel stages, respectively. Similarly, I1800 supplied 900 m3 ha−1 for each stage of anthesis and the kernel stage.
Lines 162-164 – 2/3 of the N was applied before sowing. This means, that a certain amount of the N could have been leached below the root zone. What was the soil type in the experimental site? Please, give soil texture data.
Response: L120-123, N60 application, 60 kg N ha-1, 40 kg N ha-1 and 20 kg N ha-1 were applied to the soil before sowing and at the budding stage. For the N120 application, 120 kg N ha-1, 80 kg N ha-1 and 40 kg N ha-1 were applied to the soil before sowing and at the budding stage.
L95-96, The soil type is dark loessial soil (Heilutu), with 50 % sand, 20 % silt, and 30 % clay.
Lines 169-173 – This paragraph is not necessary. The readers know what a split-plot design is. Give them some credit.
Response: The author deleted this part.
Lines 181-190 – You determined soil water content gravimetrically (eq. 4). To what depth did you take soil samples?
Response: L144-146, The soil water content (SWC) in the 0-160 cm layer was measured every 20 cm before sowing and after harvesting, and 5 points were randomly selected in each plot.
Lines 193-195 – Eta is usually used for indicating evapotranspiration, measured by a class A pan or calculated using the Penman or Penman-Monteith equation. The use of Eta in this case is confusing. I suggest using another term.
Response: The authors substitute evapotranspiration with water consumption.
Eq. 9 – Similarly to equations 6, 7 and 8, eq. 9 calculates NUE.
Response: ANUE was agronomy N use efficiency and NPFP was nitrogen partial factor productivity.
Please, describe how yield and yield components were assessed.
Response: L153-154, At the end of the growing season, the plants in each plot were hand harvested for grain yield.
- Results
Table 2 – In eq. 9 you define NFP (N fertilizer productivity) as GY/N application rate and the units are kg kg-1. In Table 2 N fertilizer productivity units are kg ha-1.
Response: The unit has been changed to kg kg-1.
Line 283 – In this line you indicate that nitrogen had insignificant effects on grain yield in both growing seasons. However, in the statistical analysis results (Table 2), you indicate that N had a very significant effect on grain yield. How do you explain this contradiction?
Lines 284-286 – your results (Table 3) do not support this statement. On the contrary. Your results give an indication (not significant though) that over-fertilization impairs production.
Tables 2 and 3 – The results of the statistical analysis in Table 2 are not always in agreement with the results presented in Table 3. Please, check your statistical analysis.
Response: An error occurred during statistical analysis, which has been corrected.
Figure 6 – Change Y axis from kg hn-2 to kg hm-2.
Response: Y axis had changed to kg ha-1.
Table 5 – Very unclearly presented. Not possible to read it.
Response: The author has abridged Table 5 to make it easier to read clearly.
- Discussion
In general, the discussion is not clearly organized. There are many repetitions, many sections belong to the conclusion section and I suggest to re-write and re-organize the discussion and make it more concise and to the point.
Response: The author re-organize the discussion and re-write some content of the discussion.
Lines 348-350 – This statement is not really correct. Your results are in agreement with those of Shangguan et al.
Response: Modified to “Our results showed that the value of WSE was 29.84-52.29%. This value is in agreement with those of Shangguan et al. [25], who reported that the WSE of the Loess Plateau is about 30-35%.”
Lines 351-352 and Eq.1 – According to eq.1, PSE is affected only by natural precipitation (rainfall), assuming that irrigation does nor occur during the fallow period. In lines 351-352 you state that PSE can be affected by irrigation. Irrigation can affect the PSE value of the subsequent season (residual water) but in this case, the term PSE is not correct and I suggest to use the term WSE (Water storage efficiency), which refers to both precipitation and irrigation.
Response: Modified “PSE” to “WSE”.
Lines 355-358 – How would N level increase PSE? Under elevated N levels, biomass production increases and therefore also plant water consumption, which leaves less water in the soil and leads to a lower PSE and SWS.
Response: L326-340, In our study, irrigation significantly increased the SWS before sowing in 2018 regardless of whether N was applied or not (Fig. 2 D). Meanwhile, N application significantly increased the SWS before sowing in 2018 under the I1800 level (Fig.e 3 B). The SWS after harvest is differ due to water-nitrogen coupling treatments, which are the result of the combined effects of precipitation, irrigation, crop production, and evapotranspiration. In our current study, irrigation significantly increased the SWS after harvest in 2017 and 2018 under the same N application rate (Figs. 2 E and F). The spatio-temporal dynamics of SWS revealed that the N application rates had a significant impact on SWS [30]. Our present study showed that N rate significantly improved the SWS before sowing in 2018 under the I1800 level; this may be related to the SWS after harvest in 2017. The N60 and N120 treatments significantly improved the SWS after-harvest in 2017 under the I1800 level compared with N0. The results demonstrated that the irrigation water was not fully utilized and that part of it was stored in the soil for absorption and utilization by the next crop.
Lines 367-379 – This paragraph describes, again, the fact that under high N application levels SWS was higher. Please, provide an explanation to this strange phenomenon. There is no consistency between the two years, apparently because of differences in precipitation, which might be an indication that the major factor determining SWS is water supply (irrigation + precipitation) and that N has a minor effect on this parameter.
Response: L340-344, The effects of high nitrogen (N120) on the SWS after harvest were different in 2017 and 2018 at I1800 level; N120 increased the SWS after harvest in 2017 and decreased the SWS after harvest in 2018. This is due to differences in precipitation during the growing season, which might be an indication that the major factor determining SWS is water supply (irrigation + precipitation) and that N has a minor effect on this parameter.
Lines 453-463 – In a previous paragraph, emphasis was given to WUE and NUE. In this paragraph, the emphasis in on grain yield. Usually, high WUE and NUE values will be obtained under deficit irrigation and relatively low N application levels, but this will always be accompanied by relatively low yields. When high yields are obtained, WUE and NUE will be lower and the optimum point is a matter of economic considerations. At the bottom line, the producer does not sell WUE or NUE but kilograms of product and therefore he would want to maximize his income at a minimum cost.
Response: The discussion of WUE and NUE was truncated to highlight grain yield.
L345-356, Generally, appropriate water and rational fertilization can increase crop yield [31]. The grain yield and biomass were increased in the current experiment by both irrigation and nitrogen application, but nitrogen had insignificant effects on grain yield in both growing seasons (Table 4). This result indicates that water was the main factor limiting oilseed flax grain yield, and nitrogen had a lower effect on oilseed flax grain yield than water. There is a close relationship between GY and water consumption, and GY is largely determined by evapotranspiration [30]. In this study, irrigation significantly increased the ET, but N had insignificant effects on ET in both growing seasons under the same irrigation level, partially explaining the similar GY among the three N application rates. Our findings showed that irrigation contributed more to GY; while N increased GY, such N benefits may be minimal under adequate water conditions due to strong water N interaction.
L3567-363, Crop yield components play a vital role in determining grain yield and these can be affected by variety, environmental conditions, and agronomic management [32]. Previous studies have demonstrated that crop yield components can be regulated by irrigation and nitrogen [33,34]. In this study, increasing irrigation under the same nitrogen level significantly increased the EC, as did increasing nitrogen application amount under the same irrigation level (Table 5). The GY increased with the increase of EC, which is in line with the previous result that EC had positive effects on GY [35].
L369-373, I1800N60 treatment achieved the highest grain yield (Table 5). These results indicate that irrigation of 1800 m3 ha−1 with 60 kg N ha−1 could meet the demand of oilseed flax for water and nitrogen in the test area and in the area with the same ecological conditions. This is consistent with the result of our previous study [33].

Reviewer 2 Report
Dear Authors,
your manuscript „Water and nitrogen coupling increased the water-nitrogen use 2 efficiency of oilseed flax” deals the effect of irrigation and N fertilization on a range of parameter (yield components, N and water use efficiency) based on a two-years experiment in Gansu Province of China.
The paper is well prepared. The measured and calculated parameters allowed a comprehnsive analysis of the effect of the factor year, irrigation amount and N level.
The results are well presented and discussed. It supports the requirement, that N fertilization should be adapted to irrigation and vice versa.
Some further comments:
Line 64/65: “a high water level resulting from irrigation can increase N emissions” => denitrification should be mentioned
Please check that the first mentioned abbreviations in the text are explained e.g. ANUE (Line 98)
Which kind of N fertilizer did you use?
Mention the “soil depth” for the measuring of SWC and bulk density in Line 188/189.
The methods for the measured parameters (e.g. yield componets) could be described more in detail.
Figure 1: Fllow season => fallow season
Unify ha or hm-2
Unify the units in the manuscript: Line 200: NFP (kg kg-1) vs Table 5 NFP (kg ha-1)
The figures are well prepard. Tables are sometimes difficult to read. Please unify the units in the table with the text.
Author Response
Dear Reviewers:
Thank you for your letter and for the reviewers’ comments concerning our manuscript entitled “Water and nitrogen coupling increased the water-nitrogen use efficiency of oilseed flax” (ID: plants-2051341). Those comments are all valuable and very helpful for revising and improving our paper, as well as the important guiding significance to our researches. We have studied comments carefully and have made correction which we hope meet with approval. Revised portion are marked in red in the paper. The main corrections in the paper and the responds to the reviewer’s comments are as flowing:
- Line 64/65: “a high water level resulting from irrigation can increase N emissions” => denitrification should be mentioned.
Response: The authors added “high soil water content promotes denitrification”.
L62-64, Despite this, high water levels cause serious environmental issues such as greenhouse gas emissions and nitrogen leaching loss below the root zone and into groundwater [12].
- Please check that the first mentioned abbreviations in the text are explained e.g. ANUE (Line 98).
Response: The author has revised the introduction and defined ANUE when it first appears.
L156, where ANUE (kg kg−1) is agronomy N use efficiency.
- Which kind of N fertilizer did you use?
Response: The authors added nitrogen fertilizer types.
L123-125, N fertilizer is urea (N content 46%), additionally, 90 kg P2O5 ha−1 was applied every plot before sowing, phosphate fertilizer is calcium superphosphate (P2O5 16%).
- Mention the “soil depth” for the measuring of SWC and bulk density in Line 188/189.
Response: The authors added the “soil depth” for the measuring of SWC and bulk density.
L140-141, where SWS (mm) is soil water storage in the 0-160 cm, and i=1, 2, 3 ... 8, SWC (%) is the soil water content, Di (g cm-3) is the soil bulk density, Hi (cm) is the soil depth.
L144-146, The soil water content (SWC) in the 0-160 cm layer was measured every 20 cm before sowing and after harvesting, and 5 points were randomly selected in each plot.
- The methods for the measured parameters (e.g. yield componets) could be described more in detail.
Response: L153-154, At the end of the growing season, the plants in each plot were hand harvested for grain yield.
- Figure 1: Fllow season => fallow season
Response: The author has revised “Fllow” to “Fallow”.
- Unify ha or hm-2
Response: The author has unified to “ha”.
- Unify the units in the manuscript: Line 200: NFP (kg kg-1) vs Table 5 NFP (kg ha-1).
Response: The author has unified to “kg kg-1”.
- The figures are well prepard. Tables are sometimes difficult to read. Please unify the units in the table with the text.
Response: The author has abridged Table 5 to make it easier to read clearly.

Round 2
Reviewer 1 Report
Accept in present form